# Development and validation of a prognostic model for mitophagy-related genes in colon adenocarcinoma: A study based on TCGA and GEO databases

**Yong Tang[☉], Haiyang Guo[☉], Lin Chen, Xiaobo Wang, Qi Chen, Lei Gou, Xiuyu Liu, Xianfei Wang[ID]***

Department of Gastroenterology, Affiliated Hospital of North Sichuan Medical College, Nanchong, Sichuan, China

☉ These authors contributed equally to this work.
* 2750853458@qq.com

**Data Availability Statement:** Data for this study are publicly available from the NIH National Cancer Institute repository (https://portal.gdc.cancer.gov/repository). Researchers should use the "Cases"

## Abstract

### Background

Mitophagy is used by eukaryotic cells to eliminate damaged mitochondria. The deregulation of this process can lead to an accumulation of dysfunctional mitochondria and is implicated in carcinogenesis and tumorigenesis. Despite increasing evidence that mitophagy is involved in the development of colon cancer, the role of mitophagy-related genes (MRGs) in colon adenocarcinoma (COAD) prognosis and treatment remains largely unknown.

### Methods

Differential analysis was used to identify differentially expressed mitophagy-related genes associated with COAD and conduct key module screening. Cox regression and least absolute shrinkage selection operator, and other analyses were used to characterize prognosis-related genes and verify the feasibility of the model. The model was tested using GEO data and a nomogram was constructed for future clinical application. The level of immune cell infiltration and immunotherapy were compared between the two groups, and sensitivity to treatment with many commonly used chemotherapeutic agents was assessed in individuals with different risk factors. Finally, qualitative reverse transcription polymerase chain reaction and western blotting were performed to assess the expression of prognosis-related MRGs.

### Results

A total of 461 differentially expressed genes were mined in COAD. Four prognostic genes, PPARGC1A, SLC6A1, EPHB2, and PPP1R17, were identified to construct a mitophagy-related gene signature. The feasibility of prognostic models was assessed using Kaplan-Meier analysis, time-dependent receiver operating characteristics, risk scores, Cox regression analysis, and principal component analysis. At 1, 3, and 5 years, the area under the receiver operating characteristic curves were 0.628, 0.678, and 0.755, respectively, for TCGA cohort, and 0.609, 0.634, and 0.640, respectively, for the GEO cohort. Drug

interface: Select "colon" in the "Primary Site" item; Select "TCGA" in the "Program" item; Select "TCGA-COAD" in the "Project" item; Select "adenomas and adenocarcinomas" in the "Disease Type" item; Continue to filter the data in the "Files" interface: Select "transcriptome profiling" in the "Data Category" item; Select "Gene Expression Quantification" in the "Data Type" item. Additional data for this study are available from the NCBI Gene Expression Omnibus repository (https://www.ncbi.nlm.nih.gov/geo/query/acc.cgi?acc=GSE17538) using accession numbers GSE17538 - GPL570. Additional data for this study are publicly available from the GeneCards database (https://www.genecards.org). Researchers should enter "mitophagy" in the search bar. See Supporting Information for more details.

**Funding:** Xianfei Wang was supported by the Sichuan Provincial Primary Health Development Research Center(no. swfz20-z-003).The funder had a role in study design, analysis, decision to publish and preparation of the manuscript.

**Competing interests:** The authors declare that they have no conflicts of interest.

sensitivity analysis found that camptothecin, paclitaxel, bleomycin, and doxorubicin were significantly different between low- and high-risk patients. The qPCR and western blotting results of clinical samples further confirmed the public database results.

## Conclusions

This study successfully constructed a mitophagy-related gene signature with significant predictive value for COAD, informing new possibilities for the treatment of this disease.

## Introduction

Globally, colon cancer has high morbidity and mortality, ranking third in cancer incidence and second in mortality [1]. The pathogenesis of colon cancer is complex and induced by sporadic, familial, hereditary and other risk factors [2]. Colon cancer is caused by one or more of chromosomal instability, CpG island methylation phenotype, and microsatellite instability [3]. Although living standards are gradually improving in China, the incidence and mortality of colon cancer there continue to increase [4]. Most (90%-95%) colon tumors have adenocarcinoma as their predominant histological type [5]. In the future, new and reliable biomarkers are expected to offer options for early diagnosis and treatment, and to aid the selection of appropriate therapeutic drugs for individual patients. Therefore, biomarkers have significant potential for prevention, diagnosis and treatment of this disease [6]. To improve survival in patients with colon adenocarcinoma (COAD), effective treatment strategies must be explored by identifying reliable prognostic biomarkers.

Mitochondria are primarily responsible for cellular energy supply, and defects in their function not only affect cellular homeostasis, bioenergetics, and redox control, but also have important implications for cell death [7]. Mitophagy is a well-conserved cellular process in which dysfunctional mitochondria are degraded and selectively eliminated. Dysregulated mitophagy results in accumulation of damaged mitochondria, with a significant role in tumor development and carcinogenesis [8]. It has been reported that mitophagy-related genes (MRGs) such as PINK1, BNIP3L and PARKIN have emerged as targets for anticancer therapy by evading drug resistance [9]. PARKIN deletion occurs frequently in colorectal cancer and is significantly associated with adenomatous colonic polyp deficiency [10]. Cancer therapy may benefit from targeting PHB2-mediated mitophagy via the PHB2-PARL-PGAM5-PINK1 axis [11]. However, no predictive signature has been constructed for COAD. Given its importance as the predominant colon tumor, there is an urgent need to comprehensively study the impact of MRGs at the genetic level on the prognosis and treatment of patients with COAD.

In this study, we analyzed RNA expression sequences and corresponding clinical data, retrieved from The Cancer Genome Atlas (TCGA) and gene expression ontology (GEO) databases, on patients with COAD. We also identified genes associated with prognosis and constructed a prognostic model by applying Cox regression and least absolute shrinkage and selection operator (LASSO) regression analyses, and finally evaluated the immune and chemotherapy response.

## Materials and methods

### 1. Data download

We obtained RNA-seq transcriptomic data and clinical information from TCGA (https://portal.gdc.cancer.gov/) database on a training cohort of 398 COAD and 39 non-oncology

patients. The GEO database GSE17538 (http://www.ncbi.nlm.nih.gov/geo/) provided RNA-seq data derived from 177 colon cancer samples and corresponding clinical information as a validation cohort. Mitophagy-related genes (n = 1899) were downloaded from GeneCards (https://www.genecards.org) prior to March 18th, 2022 (S1 Table).

## 2. Difference analysis and enrichment analysis

We screened COAD differential genes from TCGA database using the Wilcoxon test in the R software limma package with a threshold of log2 (FC) > 1, and a false discovery rate of P < 0.05. The Pheatmap package was used to create heatmaps and volcano maps. Gene ontology (GO) and Kyoto Encyclopedia of Genes and Genomes (KEGG) pathway enrichment analyses were conducted in R (org.Hs.eg.db,clusterProfiler,ggplot,enrichplot) to assess the associated biological functions and pathways. Significance thresholds for output categories were set at p < 0.05 and q < 0.05.

## 3. PPI network construction and key module screening

We utilized the STRING (https://cn.string-db.org/) database to study interactions between the related proteins of these differentially expressed MRGs (DEMRGs). Cytoscape (version 3.9.0) was used to build the protein-protein interaction (PPI) network and the plugin MCODE was used to obtain the important molecular modules in the network.

## 4. Identification of prognosis-related MRGs and construction of a prognosis model

Using the survival package in R, Cox regression analysis was conducted to find prognosis-related MRGs among DEMRGs with a P value < 0.01. Error caused by overfitting was eliminated using LASSO. Finally, multivariate Cox regression analysis was used to construct a prognostic model.

## 5. Validation of the prognostic model

A risk score represents the prognostic risk for each patient with COAD. Based on the coefficients and expression values of the genes in the prognostic model, the risk scores of the patients in the testing and training groups were calculated. The model gene risk score was calculated as: coefficient of gene a* expression of gene a+ coefficient of gene b* expression of gene b+...+ coefficient of gene i* expression of gene i (where a, b and i represent genes). Based on the median risk score for each cohort, we divided patients with COAD into high- and low-risk groups, and then plotted Kaplan-Meier survival curves to determine whether survival differed significantly between the two groups. Moreover, receiver operating characteristic (ROC) curves were generated to determine the predictive model's accuracy. Area under the curve (AUC) > 0.75 is considered to have excellent predictive value, while > 0.60 is considered acceptable [12]. Finally, the likelihood of overall survival (OS) was predicted by nomogram analysis using the R software RMS package. Prognosis-related MRGs were identified as described above and validated using the external validation cohort GSE17538. Inclusion criteria for the external validation cohort were at least 100 sample size and complete data including age, sex, tumor stage, overall time, and survival status.

## 6. Assessment of immune/chemotherapy response

The CIBERSORT algorithm was used to assess the fraction of infiltrating immune cells. The R limma package was used to compare this fraction between high- and low-risk groups. Tumor

Immune Dysfunction and Exclusion (TIDE) (http://tide.dfci.harvard.edu/) was used to predict the response of the two groups to immune checkpoint blockade. In addition, based on cancer drug sensitivity genomics, the half inhibitory concentration (IC50) was determined in each risk group using ggplot2 of pRRophetic and R software ($p < 0.05$).

## 7. Clinical sample collection

Cancer and adjacent tissue samples were collected during July 20–25, 2022 from ten COAD patients awaiting cancer surgery at the Affiliated Hospital of North Sichuan Medical College. The patient demographic and clinical information is shown in Table 1. After tissue collection, specimen information was kept strictly confidential. None of the patients with COAD had undergone anticancer treatment such as chemotherapy or radiotherapy prior to surgery. All tissue specimens were collected within 30 minutes of surgery, placed in liquid nitrogen, and stored at -80˚C. The procedures were approved by the Ethics Committee of the Affiliated Hospital of North Sichuan Medical College and prior to the collection of tissue samples signed informed consent was obtained from each patient.

## 8. RNA extraction and qRT-PCR

We obtained total RNA from COAD and paracancerous tissue using Trizol reagent (Thermo Scientific, USA). The extracted total RNA was reverse transcribed to cDNA using HiScript® III RT SuperMix for qPCR (Vazyme Biotech Co., Ltd, China). Finally, quantitative reverse transcription polymerase chain reaction (qRT-PCR) was performed in LightCycler 96 (Roche, Switzerland) using ChamQ Universal SYBR qPCR Master Mix (Vazyme Biotech Co., Ltd, China). GAPDH was used as an internal reference. RNA levels in COAD and paracancerous colon samples were derived using the 2-ΔΔCt algorithm. The primer sequences are listed in Table 2.

## 9. Western blot analysis

Eight tissue specimen pairs were cut into pieces and placed in 1% PMSF and RIPA tissue/cell lysate (Solarbio, China) on ice for 30 minutes. The lysed tissue was then centrifuged at 12,000 rpm at 4˚C for 15 minutes. The supernatant was removed and the protein concentration was determined using the BCA Protein Assay Kit (Beyotime, China). SDS-PAGE protein loading buffer (5X) (Beyotime, China) was added and the mixture was boiled for 10 minutes. The protein (30 μg) was added to a prepared 12% SDS-PAGE gel for electrophoretic separation and transferred to a 0.45 μm PVDF membrane (Amersham Hybond, GE Healthcare). The

**Table 1. Demographic and clinical characteristics of the patients.**

| Number | Gender | Age | TNM |
|---|---|---|---|
| patient1 | Male | 80 | T3N0M0 |
| patient2 | Female | 72 | T3N1M0 |
| patient3 | Male | 67 | T2N0M0 |
| patient4 | Female | 56 | T3N0M0 |
| patient5 | Male | 53 | T4N1M0 |
| patient6 | Male | 54 | T4aN0M0 |
| patient7 | Female | 51 | T2N0M0 |
| patient8 | Male | 78 | T2N1M0 |
| patient9 | Female | 61 | T4aN0M0 |
| patient10 | Male | 80 | T3aN2M0 |

**Table 2. Primer sequences used for quantitative real-time PCR.**

| Primer | Sequence (5′–3′) |
|---|---|
| GAPDH-F | GGAGTCCACTGGCGTCTTCA |
| GAPDH-R | GTCATGAGTCCTTCCACGATACC |
| PPARGC1A-F | TCTGAGTCTGTATGGAGTGACAT |
| PPARGC1A-R | CCAAGTCGTTCACATCTAGTTCA |
| SLC6A1-F | GGGTATGGAAGCTGGCTCCTA |
| SLC6A1-R | AGGGGTTGTCGCACTGTTTC |
| EPHB2-F | AGAAACGCTAATGGACTCCACT |
| EPHB2-R | GTGCGGATCGTGTTCATGTT |
| PPP1R17-F | CGTTGCAGCCACTTAGATGAT |
| PPP1R17-R | GGTCTGACTCAACATTCAGGGT |

membrane was blocked with a rapid blocking solution for 30 minutes and incubated with EPHB2 (1:1000, AF5246, Affinity), SLC6A1 (1:1000, DF4510, Affinity), PPP1R17 (1:1000, PA5-61599, Invitrogen), PPARGC1A (1:1000, FNab06351, FineTest), or GAPDH (1:1000, ab181602; Abcam) antibodies overnight on a 4°C shaker and washed three times with TBST (0.1% Tween-20) for 10 minutes. The membrane was then incubated with goat anti-rabbit IgG (H + L) HRP (1:3000, S0001; affinity) for 1 hour, washed, and developed using BeyoECL Star (Beyotime, China).

## 10. Statistical analysis

R software (version 4.1.2) (https://www.r-project.org/) and GraphPad Prism 8.0.1 (GraphPad Software Inc., La Jolla, USA) were used for all statistical analyses. The qRT-PCR results of the 10 pairs of tissues were compared using a paired t-test. Significance levels were set at $p < 0.05$ unless otherwise stated (*$p < 0.05$, **$p < 0.01$, **$p < 0.001$).

## 11. Ethics statement

This study was approved by the Ethics Committee of the Affiliated Hospital of North Sichuan Medical College (2022ER240-1) and performed according to the declaration of Helsinki. Informed consent forms were signed by all patients.

## Results

### 1.DEMRGs

A total of 461 DEMRGs were identified between COAD and noncancerous tissue, including 290 up-regulated and 171 down-regulated MRGs (see Fig 1).

### 2.GO and KEGG pathway enrichment analysis of DEMRGs

Biological Process (BP) analysis found that up-regulated MRGs were enriched in ribonucleo-protein complex biogenesis, DNA conformation changes, chromatin remodeling, and nucleosome assembly. Down-regulated MRGs were significantly enriched in cellular response to chemical stress, response to oxidative stress, response to reactive oxygen species, and muscle system process. In Cellular Component (CC) analysis, up-regulated MRGs were significantly enriched in protein-DNA complex, nucleosome, and DNA packaging complex, while down-regulated MRGs were significantly enriched in blood microparticles, mitochondrial outer membranes, and presynaptic membranes. In the Molecular Function (MF) analysis, the up-

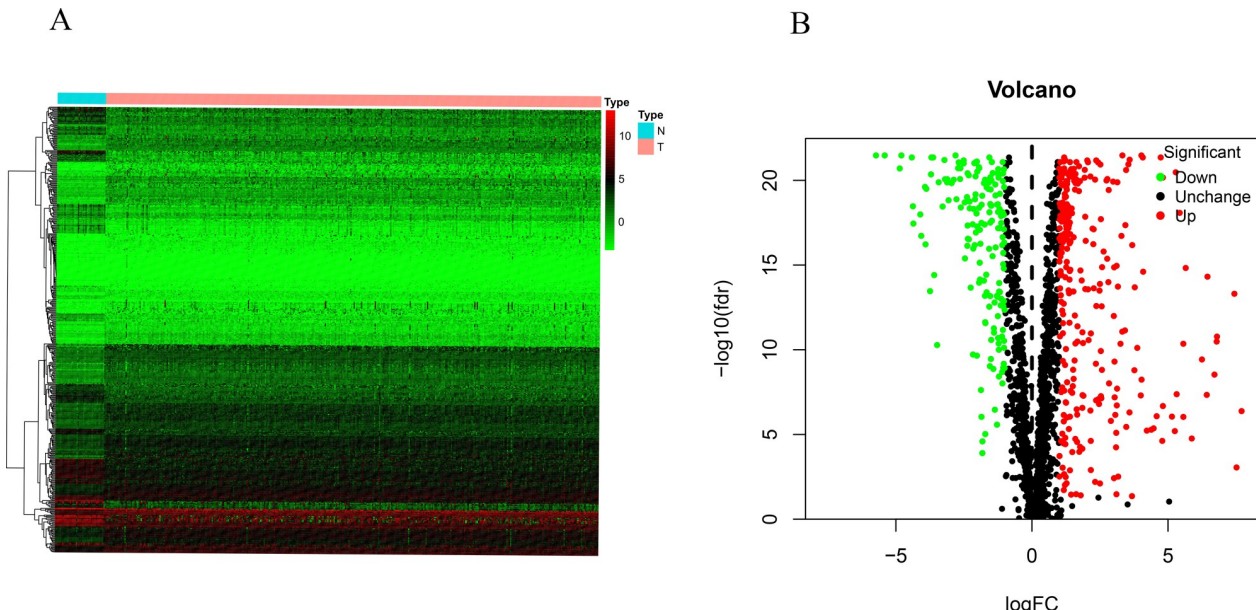

**Fig 1. Heatmap and (B) volcano plot of differentially expressed MRGs (DEMRGs).** Red and green show high and low gene expression levels, respectively; black indicates no significant difference in expression level.

regulated MRGs were significantly enriched in protein heterodimerization activity, helicase activity, and mRNA 5'-UTR binding. There was also a significant enrichment of actin filament binding, actin binding and ubiquitin-like protein ligase binding in the down-regulated MRGs (Fig 2A and 2B).

A KEGG pathway enrichment analysis of DEMRGs revealed that up-regulated MRGs were enriched in neutrophil extracellular trap formation, alcoholism, and systemic lupus erythematosus, while down-regulated MRGs were enriched in the MAPK signaling pathway, Parkinson's disease, and focal adhesion (Fig 2C and 2D).

## 3.PPI network construction and key module screening

The PPI network included 433 nodes and 2327 edges. Further analysis identified the three most important modules, with 19 nodes and 152 edges in module 1, 21 nodes and 158 edges in module 2, and 19 nodes and 66 edges in module 3. Functional enrichment analysis showed that module 1 genes were enriched for ribosome biogenesis, rRNA processing, ribosome biogenesis in eukaryotes, and mismatch repair. The genes of module 2 were expressed in mitotic cell cycle phase transition, cell cycle, chromosome segregation. In addition, Human T-cell leukemia virus 1 infection was significantly enriched. Module 3 genes were enriched in wound healing, regulation of phosphatidylinositol 3-kinase signaling, proteoglycans in cancer, and the PI3K-Akt signaling pathway (Fig 3).

## 4.Selection of prognosis-relevant MRGs

We identified 340 DEMRGs in the PPI network of the TCGA cohort and in the GEO cohort. Univariate Cox regression analysis identified nine central MRGs associated with prognosis. A 4-gene model (PPARGC1A, SLC6A1, EPHB2, PPP1R17) was generated by multivariate Cox regression analysis (Fig 4).

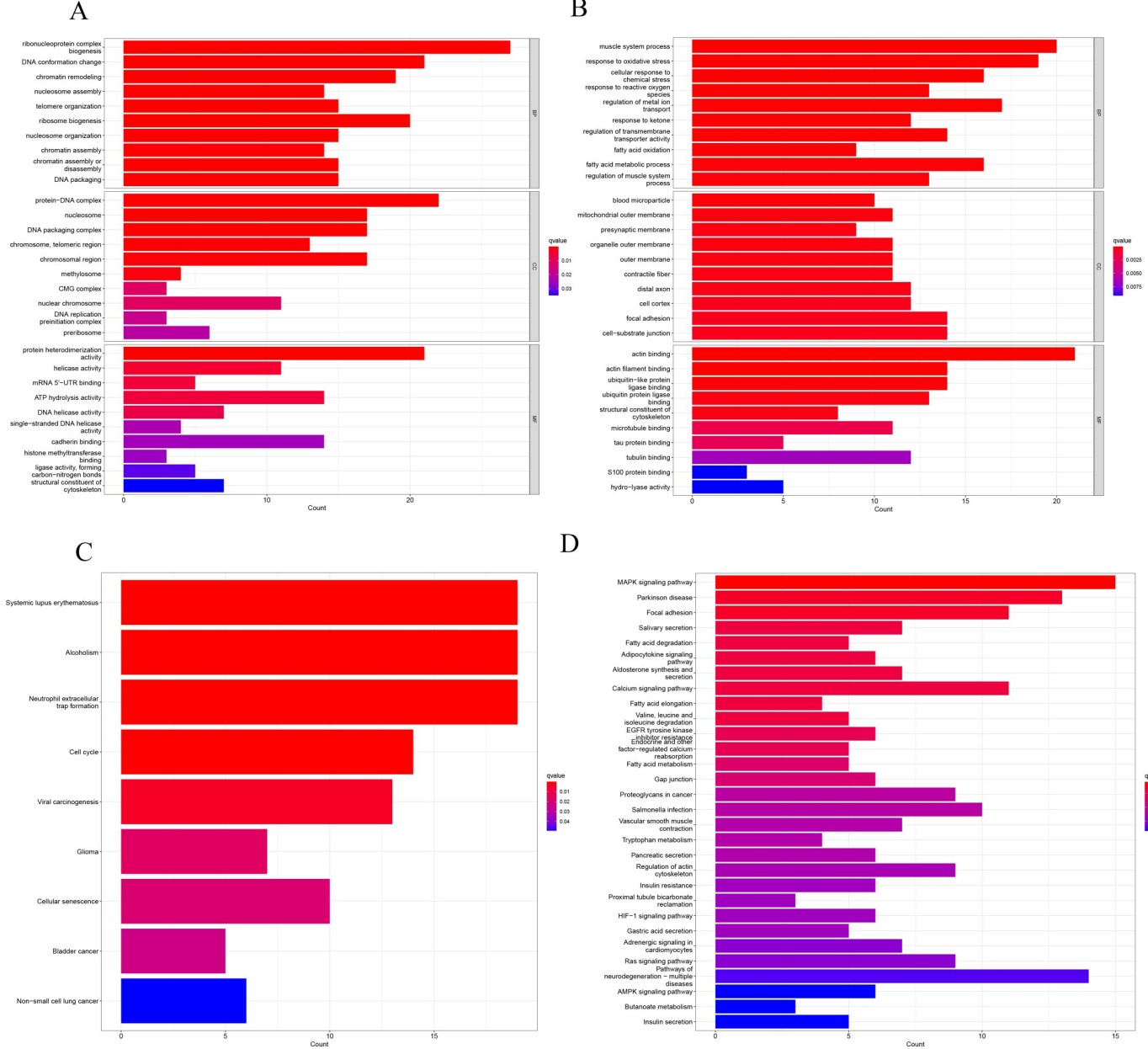

**Fig 2.** (A,B) Histograms showing results of the GO enrichment analysis, (C,D) Histograms showing results of the KEGG enrichment analysis.

## 5.Construction of an MRG-related risk-scoring model

The risk score for COAD was derived using the four previously obtained prognostic MRGs and the following formula: (-0.551 * PPARGC1A Exp) + (1.769 * SLC6A1 Exp) + (-0.417 * EPHB2 Exp) + (1.258*PPP1R17Exp). SLC6A1 and PPP1R17 indicated risk (hazard ratio (HR) > 1) whereas PPARGC1A and EPHB2 indicated protective factors (HR < 1) (Table 3). Based on the median risk score of the training group, the 379 COAD patients were categorized as low-risk (n = 190) and high-risk groups (n = 189). Survival analysis revealed significantly longer OS in the low- than high-risk group (p = 2.639e-04; Fig 5A). ROC analysis of the MRG risk score model showed AUCs of 0.628, 0.678, and 0.755 at 1, 3, and 5 years, respectively (Fig 5B).

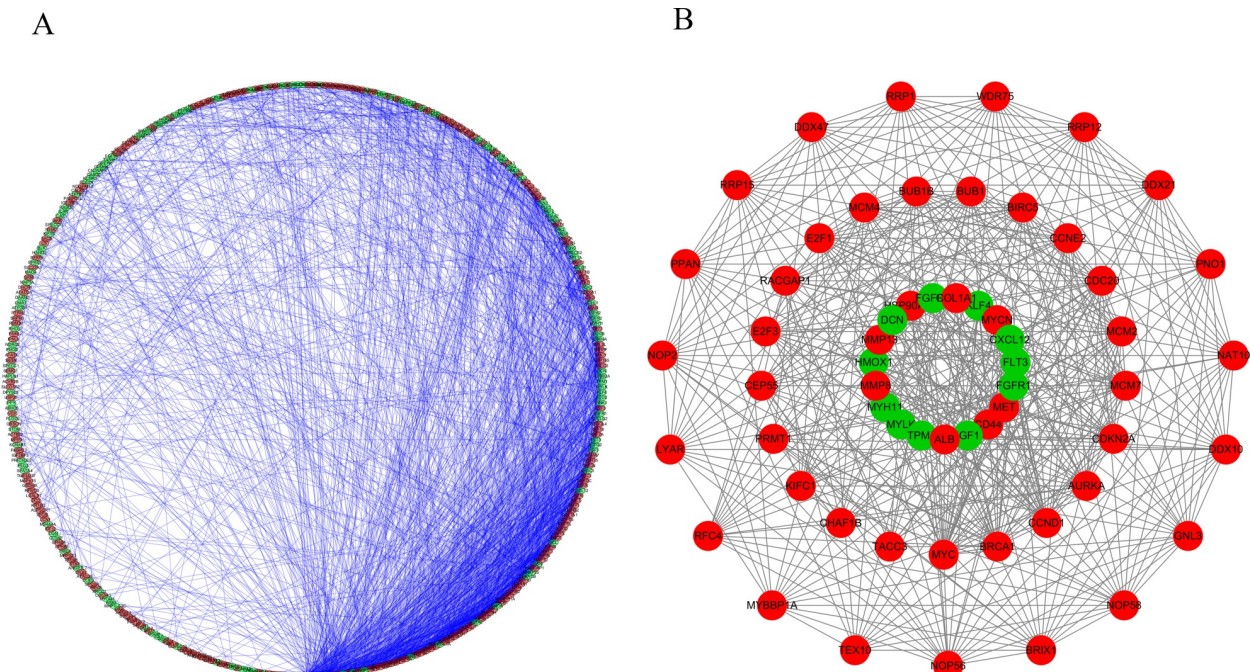

**Fig 3. Protein-protein interaction (PPI) network and module analysis.** (A) PPI network of DEMRGs. (B) Genes in the first three key modules are shown as rings from outermost (module 1) to innermost (module 3). Green and red represent down-regulated and up-regulated MRGs, respectively.

Fig 5C, 5D and 5E show patient survival status, expression heatmaps and risk scores, respectively, in high- and low-risk groups. A model to determine whether these four MRGs had predictive value in the GEO database is shown in Fig 6A–6E. Results of the PCA analysis showed that the model could distinguish the patients with COAD based on their level of risk (Fig 6F and 6G). These results indicate that our prognostic model had good specificity and sensitivity.

## 6. Factors Independently Associated with OS and the Construction of a Nomogram Model

Clinical data and risk scores were combined for patients with COAD. Univariate and multivariate (Fig 7A and 7B respectively) Cox regression analyses of data in the training group showed that age, tumor stage, and risk score were all associated with survival. In the GEO test group, univariate and multivariate Cox regression analyses showed that tumor stage and risk score were significantly associated with survival (Fig 7C and 7D respectively). A nomogram based on the four key MRGs and clinical information (gender, age, and stage) revealed 1-, 3- and 5-year survival rates of 0.913, 0.813 and 0.676 (Fig 7E and 7F).

## 7. Assessment of immune/chemotherapy response

As described earlier, patients were categorized into high- and low-risk groups. Fig 8 shows the composition of immune cells (A) and the correlations between those cells (B). We found that the fraction of macrophages M0, T cells CD4 memory resting, and dendritic cells resting differed between the two groups (Fig 8C). The TIDE score was used to evaluate the response to immunotherapy in the high- and low-risk groups. The higher the score, the greater the potential for immune escape, and the worse the effect of immunotherapy. Treatment led to improved outcomes that did not differ significantly between the two groups (Fig 8D).

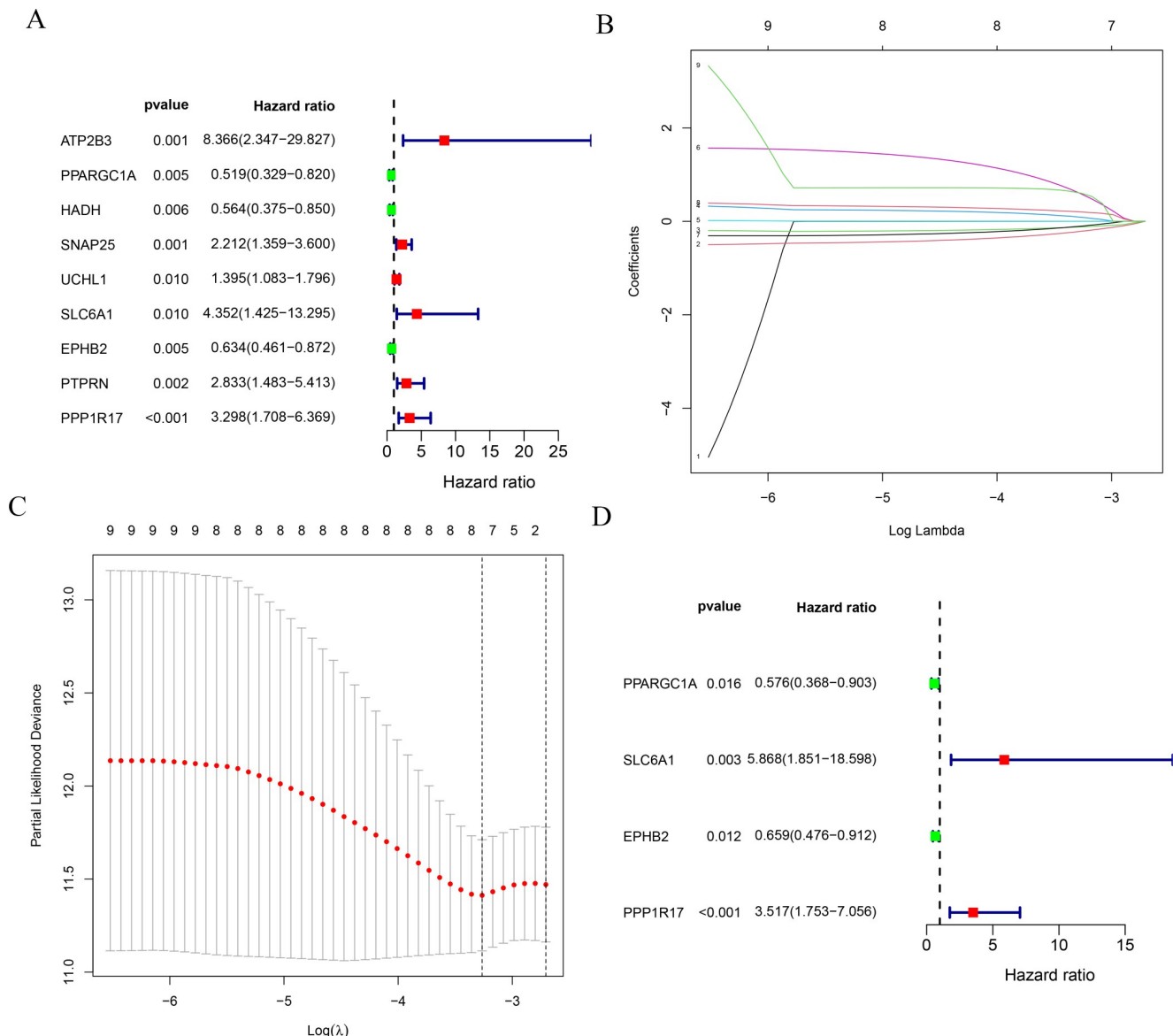

**Fig 4. Selection of prognosis-related MRGs in the training group.** (A) Results of univariate Cox regression analysis. (B and C) Results of least absolute shrinkage and selection operator (LASSO) regression analysis. (D) Mitophagy-related genes (MRGs) most relevant to prognosis identified by multivariate Cox regression analysis.

The results of drug sensitivity analysis showed that camptothecin, paclitaxel, bleomycin and doxorubicin were significantly different between low- and high-risk patients, which may provide a reference for chemotherapy in patients with COAD (Fig 9).

**Table 3. Four gene signature selected by multivariate Cox regression.**

| id | coef | HR | HR.95L | HR.95H | pvalue |
|---|---|---|---|---|---|
| PPARGC1A | -0.551444 | 0.5761173 | 0.3676993 | 0.902669871 | 0.0160883 |
| SLC6A1 | 1.7694856 | 5.8678343 | 1.8513406 | 18.59813372 | 0.0026435 |
| EPHB2 | -0.417063 | 0.6589796 | 0.4762228 | 0.911871764 | 0.0118473 |
| PPP1R17 | 1.2575101 | 3.5166543 | 1.7525706 | 7.05641035 | 0.0004016 |

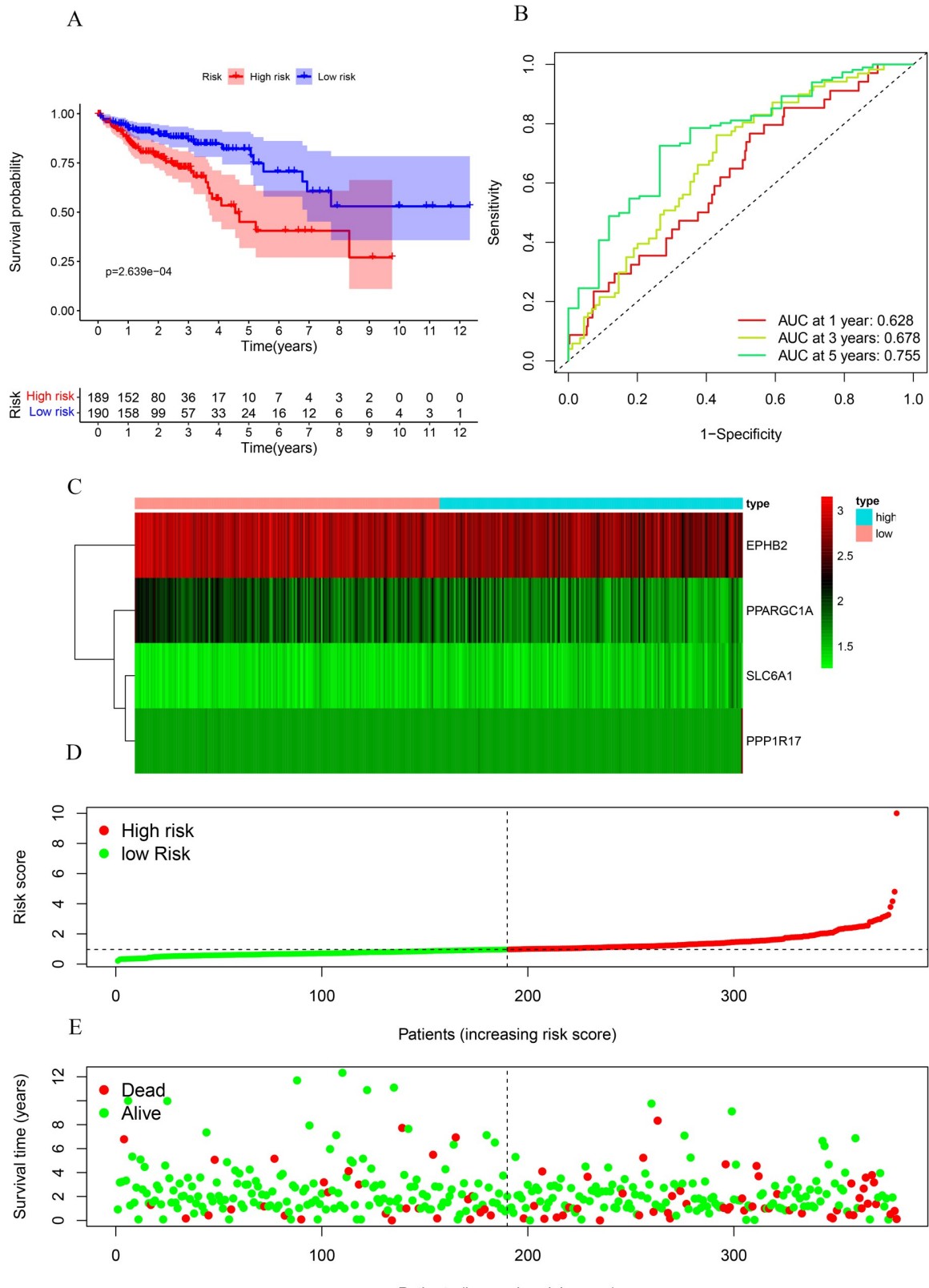

**Fig 5. Risk score analysis of the four-gene prognostic model in the TCGA cohort.** (A) Survival curve for low- and high-risk groups. (B) ROC curves for prediction of overall survival based on the risk score. (C,D,E) Expression heat map, risk score distribution, and survival status.

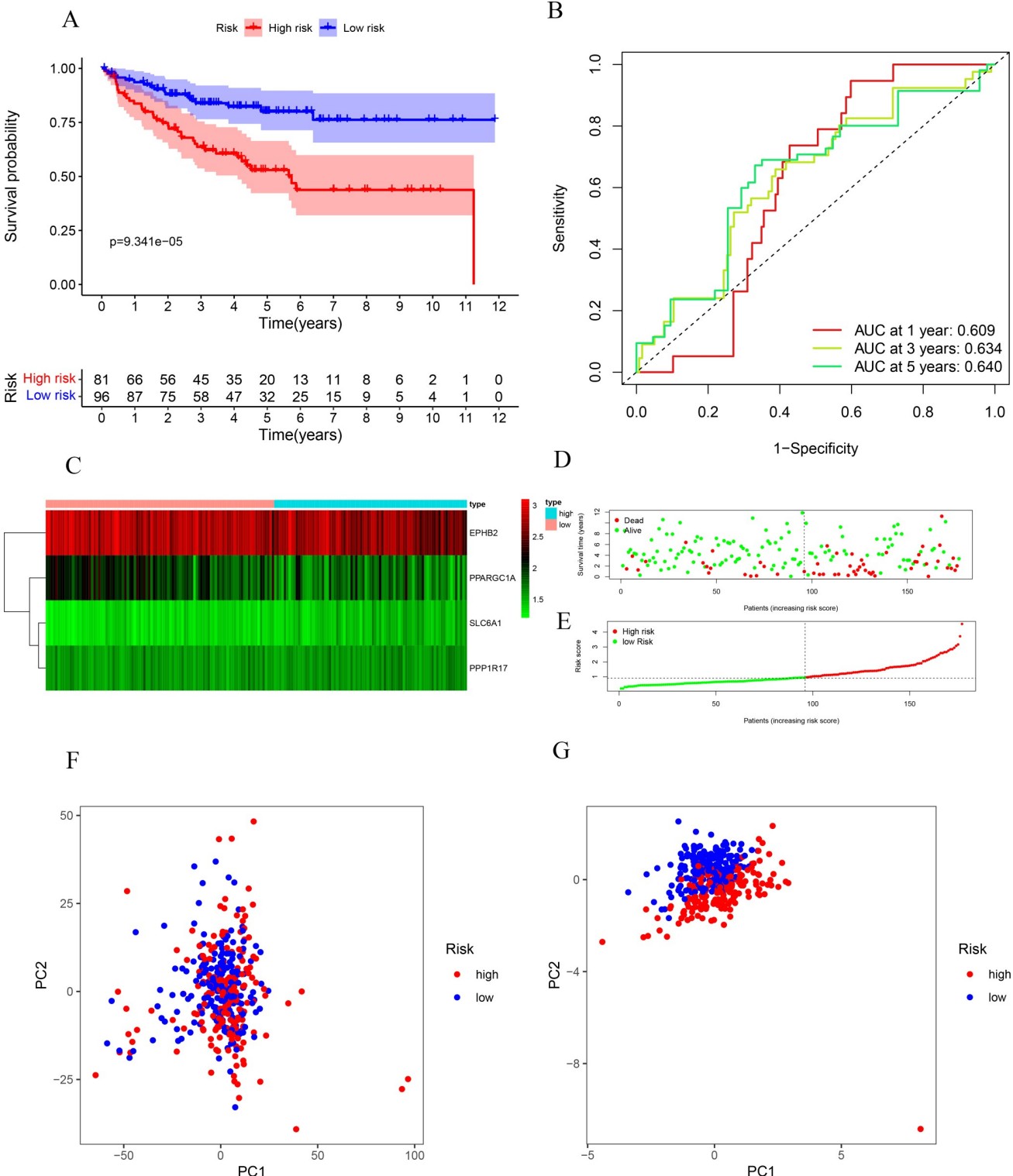

**Fig 6. Risk score analysis of the four-gene prognostic model in the GEO cohort.** (A) Survival curve for low-risk and high-risk groups. (B) ROC curves for prediction of overall survival (OS) based on the risk score. (C,D,E) Expression heat map, risk score distribution, and survival status.PCA analysis of MRGs (F) and model genes (G).

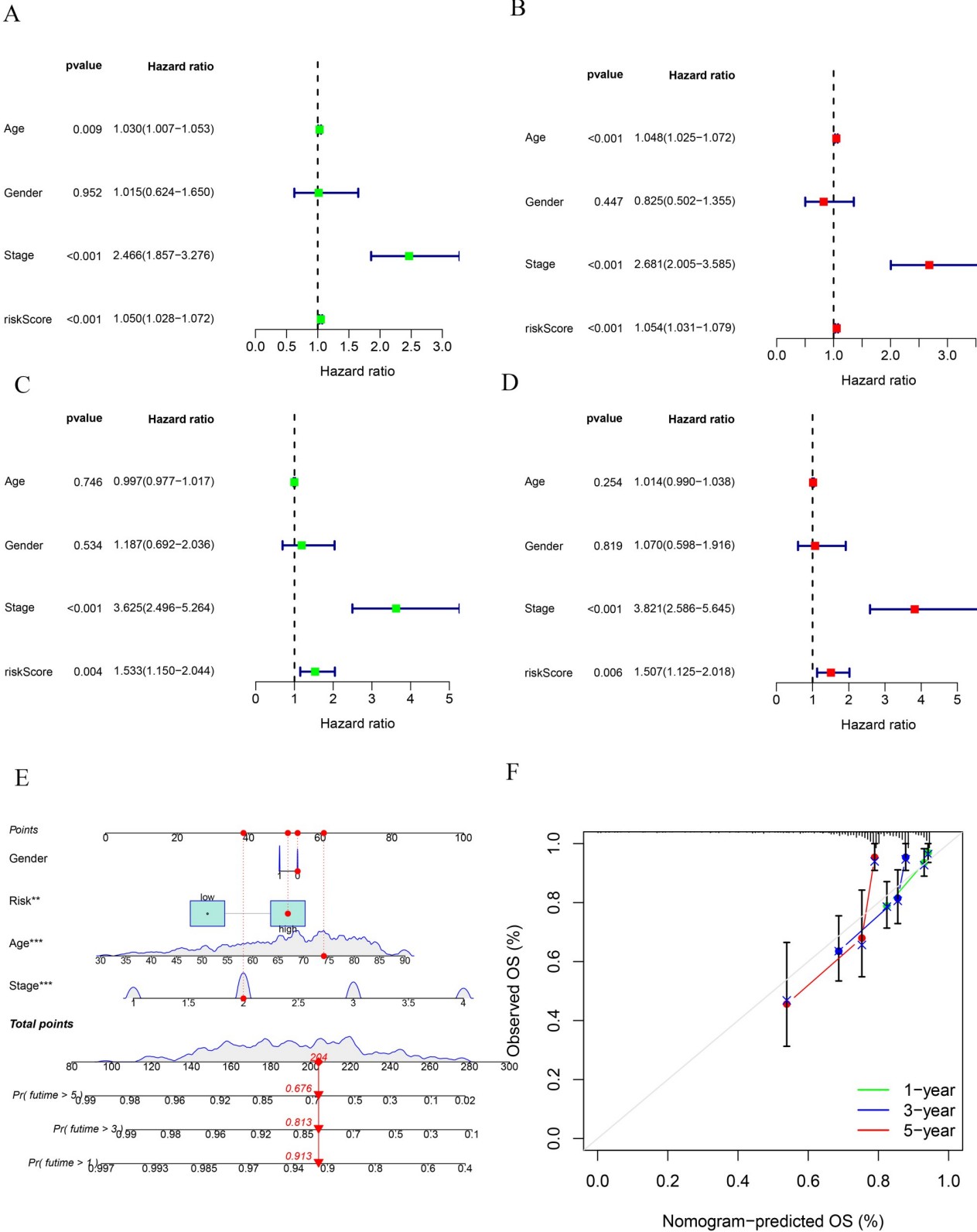

**Fig 7. Results of univariate and multivariate Cox analyses.** Univariate (A) and multivariate (B) independent prognostic analysis of training group in TCGA. Univariate (C) and multivariate (D) independent prognostic analysis of test group in GEO. Nomogram (E) for prediction of the 1-, 3-, and 5-year OS of patients with COAD in TCGA cohort (each variable is given a score, and an overall score is derived to predict a 1-, 3-, and 5-year

probability of survival). (F) Calibration curves for nomogram (closer proximity of the calibration curve to the gray line indicates higheraccuracy of the nomogram).

### 8.qRT-PCR validation of four prognostic MRGs

QRT-PCR was used to verify the expression of the four MRGs in the prognostic model, and the results were consistent with TCGA analysis of gene expression. However, the accuracy of the model was verified. Compared with paracancerous samples, EPHB2, SLC6A1, and PPP1R17 were up-regulated in the COAD samples, while PPARGC1A was down-regulated (Fig 10 and Table 4).

### 9.Protein expression of the four prognostic MRGs

Expression of the four MRGs was verified in the prognostic model and the results were consistent with TCGA gene expression analysis. However, the accuracy of the model was verified (Fig 11).

## Discussion

Colon cancer has a high recurrence rate, short survival and poor quality of life [13], so the search for specific and sensitive biomarkers is important [14]. While the role of mitophagy in carcinogenesis, tumor progression and anticancer therapy has been studied [8,15], the prognostic value of MRGs in COAD has not been systematically investigated. The risk model developed here can be validated by comparing OS and ROC curves between the training and validation sets (TCGA and GEO cohorts). Development of a mitophagy-related model that predicts prognosis and survival in patients with COAD has been facilitated by advances in sequencing technologies.

Our modular analysis of PPI networks showed that DEMRGs were mainly enriched in ribosome biogenesis, rRNA processing, ribosome biogenesis in eukaryotes, mismatch repair, cell cycle, chromosome segregation and mitotic cell cycle phase transition. The nucleolus has important roles in normal function and in tumors, as the site of the interphase ribosomal gene and of ribosome biogenesis, with links to cell proliferation [16]. The link between the proliferative growth of colorectal cancer cells and ribosome biogenesis has been highlighted as a key driver of tumorigenesis [17]. rRNA processing optimizes ribosome biosynthesis and various functions performed by mature ribosomes by fine-tuning ribosomal structure, and rRNA processing changes affect translation fidelity and translation initiation patterns of key cancer genes [18,19]. The essential components of mismatch repair are the MutS and MutL homologs, and loss of function of the mismatch repair gene results in hypermutation and high microsatellite instability. Studies have shown that sporadic cases of mismatch repair-deficient, high-grade microsatellite instability tumors are most common in colorectal cancer, with an incidence of approximately 15% [20–22]. These findings indicate that MRGs may regulate various biological processes to affect the growth of tumor cells.

In the present study, a model was established using Cox and LASSO regression to identify four key MRGs related to the prognosis of patients with COAD, including PPARGC1A, SLC6A1, EPHB2 and PPP1R17. PPARGC1A regulates energy metabolism, and the protein encoded by this gene is a transcriptional coactivator. Tumor cells enhance oxidative phosphorylation, mitochondrial biosynthesis, and oxygen consumption rate through this gene [23]. Studies have shown that PPARGC1A promotes breast cancer metastasis and is upregulated and promotes lung cancer metastasis [24,25]. Meanwhile, consistent with the present PCR and

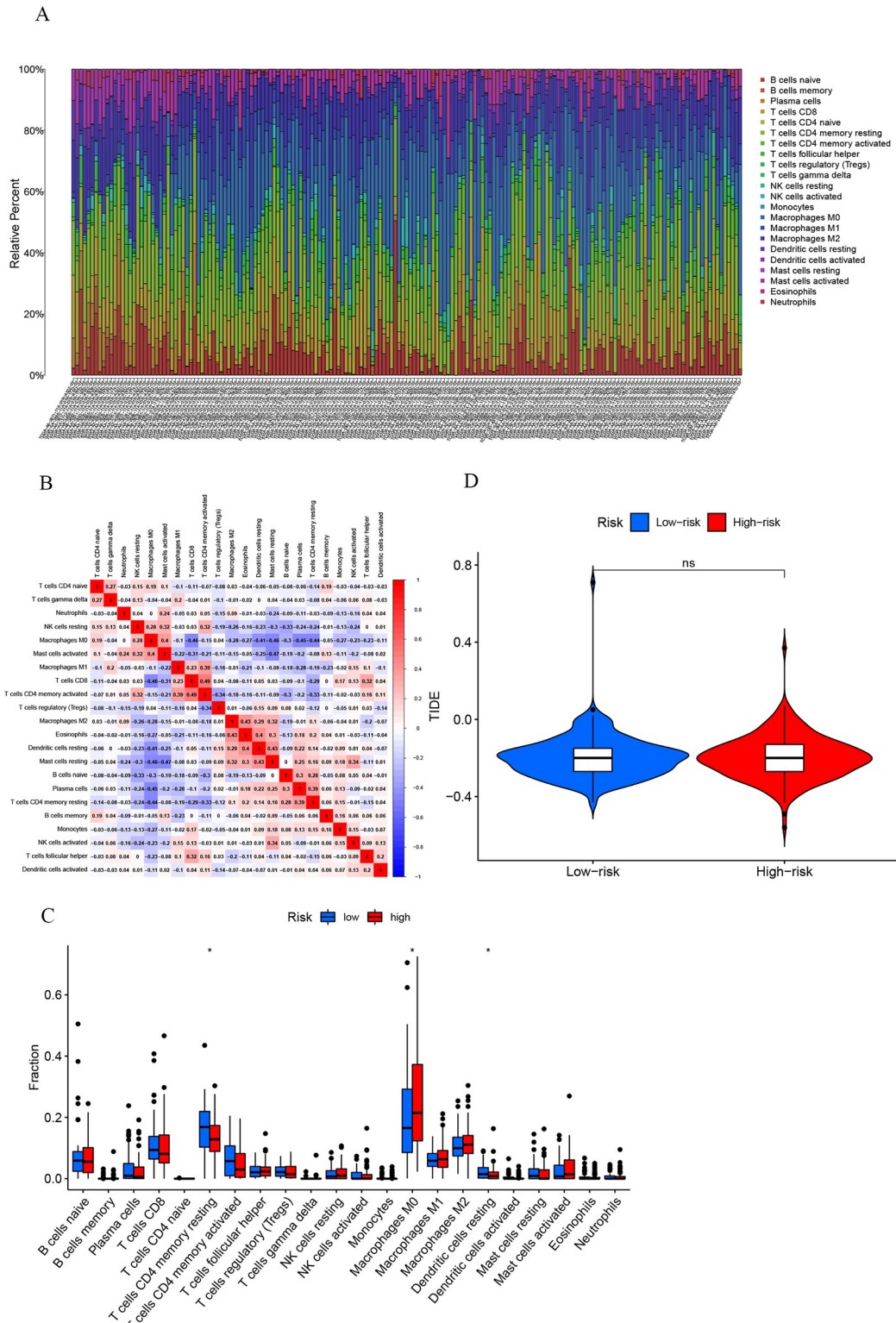

**Fig 8.** (A) Estimated scores of 22 immune cell subtypes. Each bar chart represents cell proportions in each patient, and different colors represent each subtype. (B) Correlation between the 22 subtypes of immune cell. (C) Results of analysis comparing the fraction of immune infiltrating cells between high- and low-risk groups. (D) Analysis of immunotherapy; the vertical coordinates represents the TIDE score, abscissa represent high- and low-risk groups (*P < 0.05, nsP > 0.05).

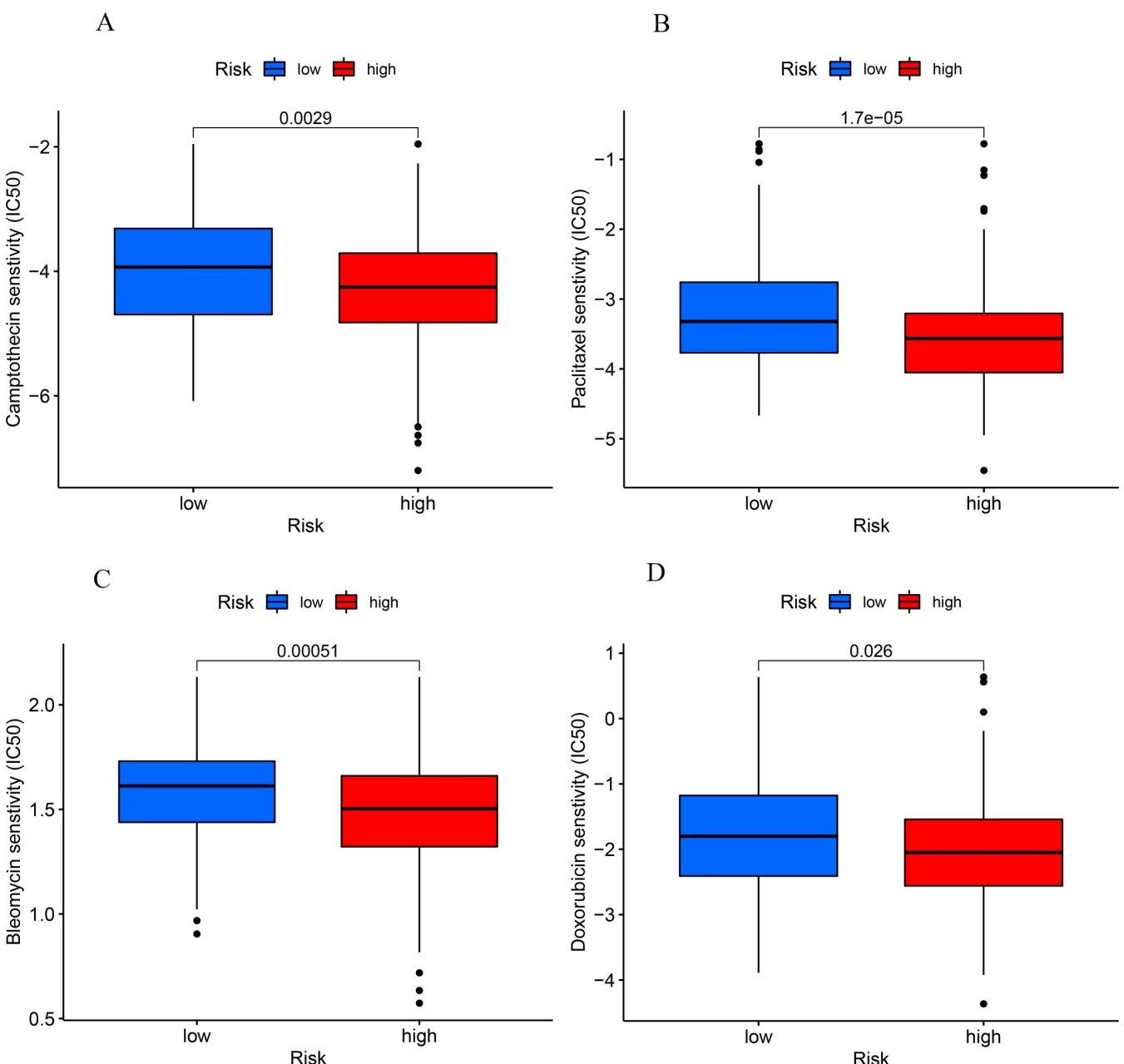

**Fig 9.** Sensitivity to camptothecin (A), paclitaxel (B), bleomycin (C) and doxorubicin (D) was significantly higher in the high- than low-risk group.

western blotting results, the expression of PPARGC1A is lower in COAD cells [26], showing that PPARGC1A plays an important role in tumors.

SLC6A1 is an important part of the GABAergic system, and its abnormal expression may be the cause of GABAergic dysfunction in different pathological conditions [27]. It has been identified as an oncogene in various human cancers and its overexpression increases the risk of prostate cancer cell proliferation, migration and invasion [28]. SLC6A1 expression is increased in ovarian cancer cells, and SLC6A1 knockdown inhibits ovarian cancer cell proliferation, migration and invasion [29]. Bioinformatics analysis shows that SLC6A1 is highly expressed in colon cancer, suggesting a new prognostic biomarker for colon cancer patients,

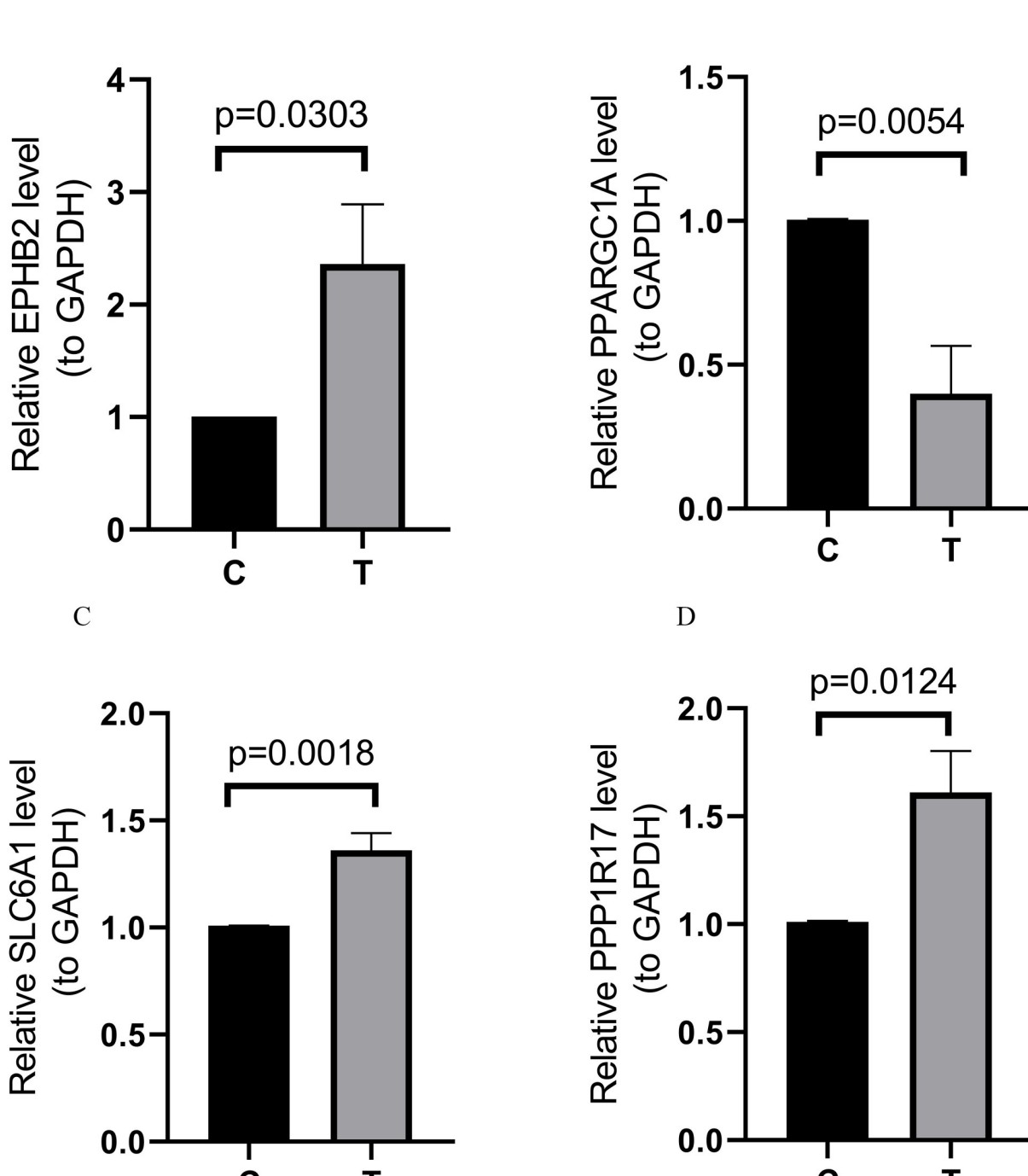

**Fig 10. Expression of MRGs in 10 sets of tissue (T, COAD tissue;C, adjacent tissue).** (A) EPHB2, (B) PPARGC1A, (C) SLC6A1, (D) PPP1R17.

pending empirical verification [30]. Our finding of SLC6A1 up-regulation in tumor tissues further suggests that SLC6A1 has potential as a new therapeutic target for COAD.

In the family of Eph receptors, EPHB2 plays a pivotal role. The gene is abnormally expressed in a variety of tumors, including gastric, bladder, and colorectal cancers, leading to

**Table 4. qPCR results.**

|  | C | T | t | df | p |
|---|---|---|---|---|---|
| EPHB2 | 1.0050±0.0034 | 2.3624±1.6729 | t = 2.568 | df = 9 | 0.0303 |
| SLC6A1 | 1.0087±0.0030 | 1.3606±0.2547 | t = 4.380 | df = 9 | 0.0018 |
| PPARGC1A | 1.0049±0.0050 | 0.3124±0.3026 | t = 3.637 | df = 9 | 0.0054 |
| PPP1R17 | 1.0116±0.0114 | 1.6114±0.6057 | t = 3.116 | df = 9 | 0.0124 |

tumor initiation and progression [31–33]. EPHB2 is overexpressed in gastric cancer cells, and EPHB2 activation enhances the malignant properties of gastric cancer cells by reducing adhesion, accelerating migration and invasive ability [31]. Correlation of low EPHB2 expression with muscle invasion in bladder cancer suggests that EPHB2 may be a key regulator of bladder cancer invasion and progression [33]. EPHB2 is highly expressed in colorectal cancer and inhibits the growth, adhesion and migration of colon cancer cells [32]. The present study found that EPHB2, with HR below 1, was a protective factor and was overexpressed in COAD cells, consistent with previous studies. The present and previous findings therefore indicate that EPHB2 is a significant factor in COAD.

PPP1R17 is a member of an open reading frame that has been found to be associated with multiple solid tumors, including colorectal [34] and lung cancer [35]. Its HR is greater than 1 in this study indicating that it is a carcinogenic factor. It is overexpressed in COAD tissues

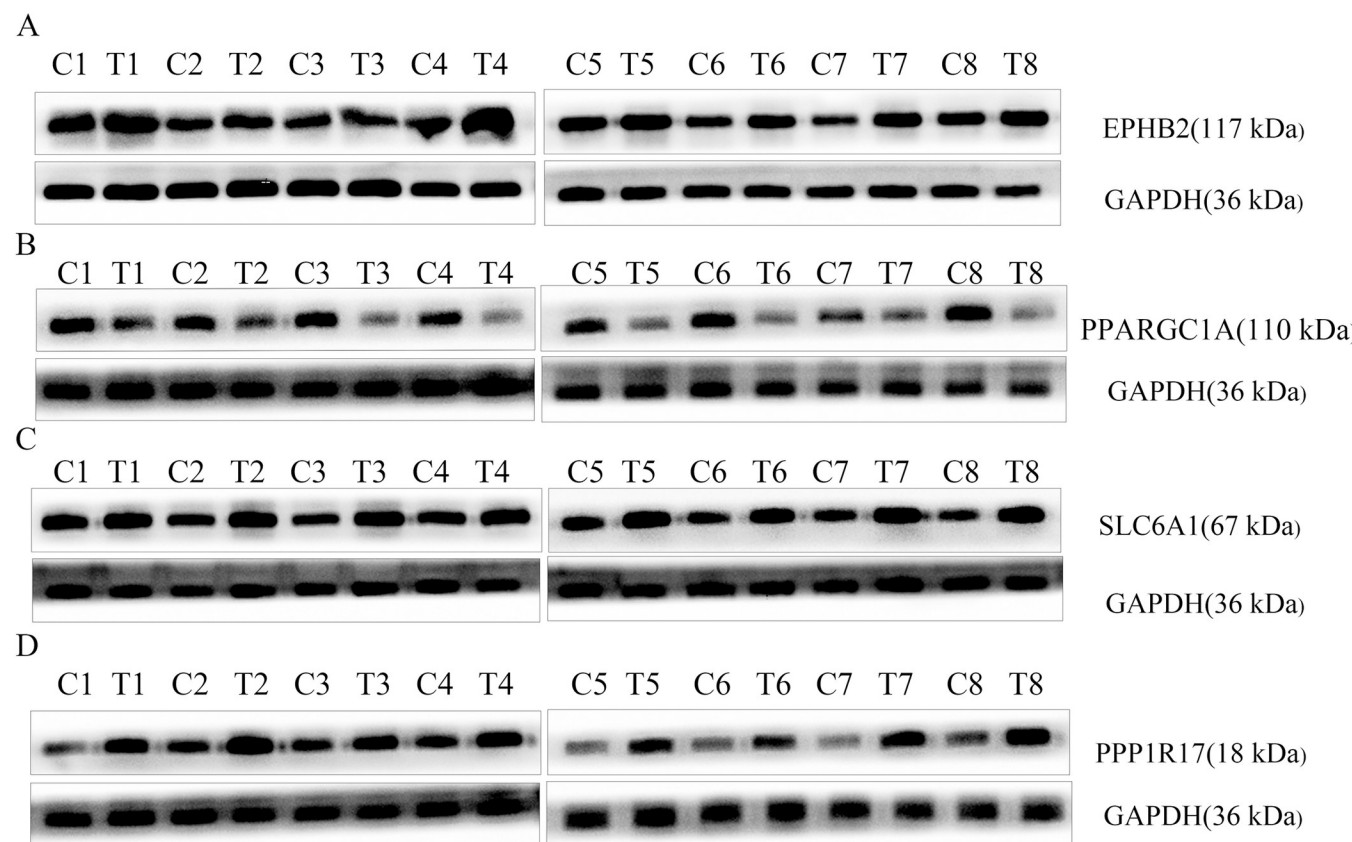

**Fig 11. Expression of the four prognostic MRGs in eight pairs of tissues was detected by western blotting (T, COAD tissue; C, adjacent tissue).** (A) EPHB2, (B) PPARGC1A, (C) SLC6A1, (D) PPP1R17.

compared to adjacent tissues, and further study of this factor is needed to understand its role in COAD and its potential role as a new therapeutic target for COAD.

Then this study evaluated the response of COAD patients to immunotherapy. It has been reported that the immune system plays an important role in the progression and treatment of cancer, and the presence of tumor immune escape protects tumor cells from the attack of the immune system [36,37]. Differentiation clusters, as markers of immune cells, such as CTLA-4, CD25, and PD-1, have been identified as potential immunotherapy targets for malignant tumors in recent years [38,39]. With the rapid development of tumor immunotherapy and bio-informatics, researchers have achieved remarkable results in obtaining new targets for tumor immunotherapy through various methods [40,41]. This study was analyzed from the genetic level, and the TIDE score showed that the high and low-risk groups had better immunotherapy effects.

The present study was limited by a small sample of clinical specimens. We hope to expand this sample in future research to determine whether MRGs have a role in molecular targeted therapy for patients with COAD. However, this study is the first to utilize MRGs to develop potentially clinically important prognostic features of COAD. It provides ideas for future research on MRGs and suggests new possibilities for treatment of COAD.

## Conclusion

The present study has identified prognosis-related MRGs, and created a prognosis-related nomogram for COAD patients. These results provide ideas and perspectives for treatment and other clinical applications of gene signatures and the nomogram.

## Supporting information

**S1 Table. 1899 Mitophagy-related genes.**
(XLSX)

## Acknowledgments

We thank the providers of the GEO and TCGA public databases and the Institute of Hepato-biliary, Pancreatic and Intestinal Diseases, North Sichuan Medical College for experimental support.

## Author Contributions

**Resources:** Lin Chen, Qi Chen, Lei Gou, Xiuyu Liu.

**Validation:** Haiyang Guo, Xiaobo Wang.

**Writing – original draft:** Yong Tang.

**Writing – review & editing:** Xianfei Wang.

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
