## [Decision Letter · Decision Letter 0]

1 Dec 2022

PONE-D-22-24625Development and validation of a prognostic model for mitophagy-related genes in colon adenocarcinoma: a study based on TCGA and GEO databases.PLOS ONE

Dear Dr. Wang,

Thank you for submitting your manuscript to PLOS ONE. After careful consideration, we feel that it has merit but does not fully meet PLOS ONE’s publication criteria as it currently stands. Therefore, we invite you to submit a revised version of the manuscript that addresses the points raised during the review process.

Specifically, please provide point-by-point response to reviewer 2's  comments. In addition, abstract needs to be rewritten to report the findings of the study in the Results (not Methods) section.

We look forward to receiving your revised manuscript.

Kind regards,

Peh Yean Cheah, Ph.D.

Academic Editor

PLOS ONE

Journal Requirements:

“This work was supported by the Sichuan Provincial Primary Health Development Research Center (no. swfz20-z-003).”

Reviewers' comments:

Reviewer's Responses to Questions

**Comments to the Author**

1. Is the manuscript technically sound, and do the data support the conclusions?

Reviewer #1: No

Reviewer #2: Yes

2. Has the statistical analysis been performed appropriately and rigorously? 

Reviewer #1: No

Reviewer #2: Yes

3. Have the authors made all data underlying the findings in their manuscript fully available?

Reviewer #1: Yes

Reviewer #2: Yes

4. Is the manuscript presented in an intelligible fashion and written in standard English?

Reviewer #1: No

Reviewer #2: No

5. Review Comments to the Author

Reviewer #1: The novelty of the manuscript is limited. The manuscript does not describe a technically sound piece of scientific research with data that supports the conclusions. This manuscript cannot be accepted for publication.

Reviewer #2: In this manuscript, the authors used the bioinformatic methods to predict some mitophagy-related genes for evaluation of survival in colon adenocarcinoma, but it did show several flaws in the design of this study.

1.The language of this manuscript needs to be radically revised.

2.In vitro or in vivo experiments should be conducted to validate the roles of mitophagy-related genes in colon adenocarcinoma.

3.In the GSE17538 dataset, the AUC was lower than 0.65. Therefore, this prognostic model is not reliable.

4.The authors used clinical samples to validate the expression of model genes, which is more convincing and can provide ethical justification?

5. Bioinformatic analysis of the study was too simple. Can authors analyze relation between prognostic models and immunotherapy.

6. PLOS authors have the option to publish the peer review history of their article (what does this mean?). If published, this will include your full peer review and any attached files.

Reviewer #1: No

Reviewer #2: No

---

## [Author Response · Author response to Decision Letter 0]

23 Jan 2023

We thank the editors and reviewers for their very helpful comments. Our revision of the reviewers ' comments has greatly improved the quality of the manuscript. The response to the reviewer 's comments is as follows :

Comment:

1.The language of this manuscript needs to be radically revised.

Response:

Thanks for your valuable suggestion. We are very sorry for our negligence.This manuscript language has been modified by a special polishing agency.

Comment:

2.In vitro or in vivo experiments should be conducted to validate the roles of mitophagy-related genes in colon adenocarcinoma.

Response:

Thanks for your valuable suggestion. The western blot experiment has been completed. Due to the novel coronavirus and laboratory conditions, other experiments could not be further completed. The EPHB2 gene in this model has been reported that its high expression reduces the migration and invasion of tumor cells, indicating that the model has a certain significance. ( Senior PV, Zhang BX, Chan ST. Loss of cell-surface receptor EphB2 is important for the growth, migration, and invasiveness of a colon cancer cell line. Int J Colorectal Dis. 2010 Jun ; 25 ( 6 ) : 687-94. doi : 10.1007 / s00384-010-0916-7. Epub 2010 Mar 26. PMID : 20339854. )

Comment:

3.In the GSE17538 dataset, the AUC was lower than 0.65. Therefore, this prognostic model is not reliable.

Response:

Many studies have reported that AUC greater than 0.6 is reliable.(

①. Wang X, Xu K, Liao X, Rao J, Huang K, Gao J, Xu G, Wang D. Construction of a survival nomogram for gastric cancer based on the cancer genome atlas of m6A-related genes. Front Genet. 2022 Aug 5;13:936658. doi: 10.3389/fgene.2022.936658. PMID: 35991573; PMCID: PMC9389082.

②. Yue Q, Zhang Y, Bai J, Duan X, Wang H. Identification of Five N6-Methylandenosine-Related ncRNA Signatures to Predict the Overall Survival of Patients with Gastric Cancer. Dis Markers. 2022 Apr 8;2022:7765900. doi: 10.1155/2022/7765900. PMID: 35774851; PMCID: PMC9239763.

③. Zhang W, Fang D, Li S, Bao X, Jiang L, Sun X. Construction and Validation of a Novel Ferroptosis-Related lncRNA Signature to Predict Prognosis in Colorectal Cancer Patients. Front Genet. 2021 Oct 28;12:709329. doi: 10.3389/fgene.2021.709329. PMID: 34777458; PMCID: PMC8581609.)

Comment:

4.The authors used clinical samples to validate the expression of model genes, which is more convincing and can provide ethical justification?

Response:

The study has passed ethical review ( 2022ER240-1 ) and has uploaded ethical review approvals.

Comment:

5.Bioinformatic analysis of the study was too simple. Can authors analyze relation between prognostic models and immunotherapy.

Response:

Thanks for your valuable suggestion. The relationship between prognostic models and immunotherapy has been analyzed.

---

## [Decision Letter · Decision Letter 1]

27 Feb 2023

PONE-D-22-24625R1

Development and validation of a prognostic model for mitophagy-related genes in colon adenocarcinoma: a study based on TCGA and GEO databases.

PLOS ONE

Dear Dr. Wang,

Thank you for submitting your manuscript to PLOS ONE. After careful consideration, we feel that it has merit but does not fully meet PLOS ONE’s publication criteria as it currently stands. Therefore, we invite you to submit a revised version of the manuscript that addresses the points raised during the review process.

Specifically, please address reviewer 1's request to summarize immunotherapy progress.

Comments from PLOS Editorial Office: We note that one or more reviewers has recommended that you cite specific previously published works. As always, we recommend that you please review and evaluate the requested works to determine whether they are relevant and should be cited. It is not a requirement to cite these works. We appreciate your attention to this request.

A marked-up copy of your manuscript that highlights changes made to the original version. You should upload this as a separate file labeled 'Revised Manuscript with Track Changes'.An unmarked version of your revised paper without tracked changes. You should upload this as a separate file labeled 'Manuscript'.If applicable, we recommend that you deposit your laboratory protocols in protocols.io to enhance the reproducibility of your results. Protocols.io assigns your protocol its own identifier (DOI) so that it can be cited independently in the future. For instructions see: https://journals.plos.org/plosone/s/submission-guidelines#loc-laboratory-protocols. Additionally, PLOS ONE offers an option for publishing peer-reviewed Lab Protocol articles, which describe protocols hosted on protocols.io. Read more information on sharing protocols at https://plos.org/protocols?utm_medium=editorial-email&utm_source=authorletters&utm_campaign=protocols.

We look forward to receiving your revised manuscript.

Kind regards,

Peh Yean Cheah, Ph.D.

Academic Editor

PLOS ONE

Journal Requirements:

Reviewers' comments:

Reviewer's Responses to Questions

**Comments to the Author**

1. If the authors have adequately addressed your comments raised in a previous round of review and you feel that this manuscript is now acceptable for publication, you may indicate that here to bypass the “Comments to the Author” section, enter your conflict of interest statement in the “Confidential to Editor” section, and submit your "Accept" recommendation.

Reviewer #1: All comments have been addressed

Reviewer #2: All comments have been addressed

2. Is the manuscript technically sound, and do the data support the conclusions?

Reviewer #1: Yes

Reviewer #2: Yes

3. Has the statistical analysis been performed appropriately and rigorously? 

Reviewer #1: Yes

Reviewer #2: Yes

4. Have the authors made all data underlying the findings in their manuscript fully available?

Reviewer #1: Yes

Reviewer #2: Yes

5. Is the manuscript presented in an intelligible fashion and written in standard English?

Reviewer #1: Yes

Reviewer #2: Yes

6. Review Comments to the Author

Reviewer #1: Most of the questions have been well addressed. However, the progress of immunotherapy need to be systematically summarized before acceptance. These reports can be referred: doi: 10.3389/fimmu.2022.1024931; doi: 10.1177/17246008211005473; doi: 10.3389/fimmu.2021.653836; doi: 10.3389/fonc.2021.822745; doi: 10.2174/0929867326666191004164041; https://doi.org/10.1186/s12863-019-0795-z.

Comments from PLOS Editorial Office: We note that one or more reviewers has recommended that you cite specific previously published works. As always, we recommend that you please review and evaluate the requested works to determine whether they are relevant and should be cited. It is not a requirement to cite these works. We appreciate your attention to this request.

Reviewer #2: The authors has solved all my problem, and this article can be accepted and published on the journal.

7. PLOS authors have the option to publish the peer review history of their article (what does this mean?). If published, this will include your full peer review and any attached files.

Reviewer #1: No

Reviewer #2: **Yes: **Yangyang Guo

---

## [Author Response · Author response to Decision Letter 1]

4 Mar 2023

Comment:

 Most of the questions have been well addressed. However, the progress of immunotherapy need to be systematically summarized before acceptance.

Response:

Thanks for your valuable suggestion. The summary of the progress of immunotherapy has been completed and added to the discussion section.

---

## [Editor Report · Decision Letter 2]

24 Mar 2023

Development and validation of a prognostic model for mitophagy-related genes in colon adenocarcinoma: a study based on TCGA and GEO databases.

PONE-D-22-24625R2

Dear Dr. Wang,

We’re pleased to inform you that your manuscript has been judged scientifically suitable for publication and will be formally accepted for publication once it meets all outstanding technical requirements.

Kind regards,

Peh Yean Cheah, Ph.D.

Academic Editor

PLOS ONE
---

## [Editor Report · Acceptance letter]

28 Mar 2023

PONE-D-22-24625R2 

Development and validation of a prognostic model for mitophagy-related genes in colon adenocarcinoma: a study based on TCGA and GEO databases. 

Dear Dr. Wang:

I'm pleased to inform you that your manuscript has been deemed suitable for publication in PLOS ONE. Congratulations! Your manuscript is now with our production department. 

Kind regards, 

on behalf of

Dr. Peh Yean Cheah 

Academic Editor

PLOS ONE